# Cyber-Physical System for Environmental Monitoring Based on Deep Learning

**DOI:** 10.3390/s21113655

**Published:** 2021-05-24

**Authors:** Íñigo Monedero, Julio Barbancho, Rafael Márquez, Juan F. Beltrán

**Affiliations:** 1Tecnología Electrónica, Escuela Politéncia Superior, Universidad de Sevilla, Calle Virgen de África 7, 41012 Sevilla, Spain; imonedero@us.es; 2Fonoteca Zoológica, Departamento de Biodiversidad y Biología Evolutiva, Museo Nacional de Ciencias Naturales (CSIC), Calle José Gutiérrez Abascal, 2, 28006 Madrid, Spain; rmarquez@mncn.csic.es; 3Departamento de Zoología, Facultad de Biología, Universidad de Sevilla, Avenida de la Reina Mercedes, s/n, 41012 Sevilla, Spain; beltran@us.es

**Keywords:** convolutional neural network, deep learning, machine learning, cyber-physical systems, passive active monitoring, Internet of Things

## Abstract

Cyber-physical systems (CPS) constitute a promising paradigm that could fit various applications. Monitoring based on the Internet of Things (IoT) has become a research area with new challenges in which to extract valuable information. This paper proposes a deep learning classification sound system for execution over CPS. This system is based on convolutional neural networks (CNNs) and is focused on the different types of vocalization of two species of anurans. CNNs, in conjunction with the use of mel-spectrograms for sounds, are shown to be an adequate tool for the classification of environmental sounds. The classification results obtained are excellent (97.53% overall accuracy) and can be considered a very promising use of the system for classifying other biological acoustic targets as well as analyzing biodiversity indices in the natural environment. The paper concludes by observing that the execution of this type of CNN, involving low-cost and reduced computing resources, are feasible for monitoring extensive natural areas. The use of CPS enables flexible and dynamic configuration and deployment of new CNN updates over remote IoT nodes.

## 1. Introduction

### 1.1. Environmental Sound Classification

Recently, international institutions such as the United Nations (UN) have paid growing attention to global strategies to coordinate the interactions among different agents (countries, companies…) around the world. These strategies [1] are focused on the development of sustainable development goals (SDGs). Special interest has been shown regarding goal 11, “sustainable cities and communities”; goal 13, “climate action”; goal 14, “life below water”; and goal 15, “life on land”. Local and regional institutions are designing roadmaps to reach these goals. This is the case for the Doñana Biological Station (DBS), a public Research Institute belonging to the Spanish Council for Scientific Research CSIC in the area of Natural Resources located in the South of Spain that focuses its research mainly on the Doñana National Park (DNP), a 135 km^2^ natural park composed of several ecosystems. The DNP is a key environment where several species of birds stop on their migration paths between Europe and Africa. These features make this space a unique enclave for analyzing several aspects of the interactions among human beings and nature.

This paper describes a project named SCENA-RBD, developed by the University of Seville in collaboration with the DBS. The main aim of the project is to design and implement a cyber-physical system (CPS) that allows distributed data processing for biological information. This CPS will constitute a useful tool for different governments (local, regional, and national) to evaluate the sustainability of an area. Several metrics [2] can be considered for this purpose. The Acoustic Entropy Index (H), which is calculated according to the Shannon information theory, indicates the richness of a specific habitat. The higher the value of a habitat’s H, the richer it is. Another common metric is the Acoustic Complexity Index (ACI), which considers the soundscape of an environment by measuring the variability of its natural audio signals in relation to audio signals with a human origin. Aligned with this idea, the Normalised Difference Soundscape Index (NDSI) is a metric that estimates the influence of human acoustic presence (anthrophony) over the biological soundscape (biophony). Based on this, a new paradigm is rising: passive acoustic monitoring (PAM). PAM is a promising research area that has recently increased in importance due to the development of low-cost wireless electronic sensors.

The CPS of the present paper consists of two stages (Figure 1). The first stage is focused on the identification of acoustic patterns. Each pattern is identified according to a specific algorithm. There could be several algorithms running at the same time depending on the target under study—biological acoustic targets such as animals or climate events or even environmental targets such as engines, gun shots, human voices, etc., could be considered in the classification. In this stage a classification procedure is executed based on previously supervised training. The second stage correlates the classification done in the previous stage with the purpose to develop biodiversity indices.

This paper is focused on the first stage of the CPS. The first step is the audio recording. Industrial devices such as the Song Meter [3] have become solid platforms for recording wildlife sounds in real-world conditions with a high quality of recording. These devices usually offer complementary software that enables a post-analysis of the recorded data. However, this sort of products has features that must be considered in an extensive sensor devices deployment. First, the price of the devices is quite high. If the area to be studied is large, a significant number of elements must be spread out. This could increase the cost of the project considerably. Although some projects are based on open hardware (e.g., AudioMoth [4]) that reduce the cost of fabrication, the possibilities they offer are limited or unstable. Second, this sort of device is designed to record audio signals on a programmable schedule. This feature involves a lack of flexibility that produces a large amount of data with no significant content. It would be desirable to ascertain certain types of events that could trigger the recording process. Third, there is no possibility of executing any preprocessing. These data are stored in raw format or with basic preprocessing. Audio waves are recorded using a standardized bandwidth (19 Hz to 19 kHz) using a sample rate of 44.1 kHz and a single channel (mono) with 16 bits per sample. A 1-minute file recorded with this configuration has a size of 5 MB. This file could be stored with or without compression. This procedure is extremely inefficient and requires large data storage. Fourth, there are no communication interfaces available. Therefore, in this sort of scenario a technician must establish a schedule for visiting every installed device to collect all the information. This task introduces an artificial effect in the measurement that contaminates the dynamics of the environment under study.

Thus, there is a need to enhance the process of environmental sound classification with a more flexible and powerful system. Not only must central processing be executed, but local and distributed processing should also be considered. Cooperative networking paradigms are suitable for this kind of scenario. In this sense, the use of Internet of Things (IoT) technology with low power consumption, low data rate, and low processing capability offers a good chance to arrive at an optimal solution. Additionally, wireless schemes of communication are desirable for scenarios in which the human presence must be minimized. In other words, the more autonomy the nodes have, the higher the quality of data obtained.

Another benefit of this approach is the scalability of the system. Covering large areas for monitoring requires deployment of a large number of devices. The use of IoT technology could ease the creation and management of the network topology. Once the nodes are spread out, they cooperate in the configuration of the network, following a spanning-tree or mesh architecture. The way they design the topology is done in an ad hoc manner. This topology is highly scalable, allowing an autonomous mechanism for network supervision.

The reduction of human physical assistance in every node of the network can be done thanks to a wireless networking cooperative feature. The network offers data transportation facilities, usually with a low data rate. In this sense, data fusion and data aggregation techniques must be considered as optimizing the information flow. Furthermore, due to the lack of infrastructure in natural environments, nodes need autonomous power supplies, commonly based on solar cells or windmills. Large area monitoring implies the use of low-cost devices. Consequently, there is a large restriction because of the capabilities of the computation resources.

At the classification stage, there is a tendency in the literature to consider semi-automated PAM analysis [5]. One of the most interesting approaches is focused on the use of modern digital image processing techniques for audio pattern identification. This strategy creates an image that represents the audio energy of a certain record in a time and frequency graph. Deep learning algorithms are run using these images [6,7,8,9]. This constitutes a data fusion technique easy to implement in low power and computation resources nodes. This way, the classification method can be done in a distributed manner, avoiding sending the audio data records through the network links, thus minimizing network power consumption. The following section describes this approach.

### 1.2. Convolutional Neural Networks in Audio Classification

The convolutional neural network (CNN) [10] is a class of deep learning neural networks oriented mainly to the classification of images. It presents a unique architecture (Figure 2) with a set of layers for feature extraction and an output layer for classification. Layers are organized in three dimensions: width, height, and depth. The neurons in one layer connect not to all the neurons in the next layer, but only to a small region of the layer’s neurons. The final output of a CNN is reduced to a single vector of probability scores, organized along the depth dimension.

Convolution is the first layer that extracts features from an input image. Convolution preserves the relationship between pixels by learning image features using small squares of input data. It carries out a linear mathematical operation that involves the multiplication of the image matrix and a filter or kernel specified in a two-dimensional array of weights.

The aim of the pooling layer function is to progressively reduce the spatial size of the representation in order to optimize the number of parameters and computations in the network. The pooling layers are typically inserted between convolutional layers. They operate on each feature map independently.

CNNs represent a major breakthrough in image recognition. They learn directly from image data, using patterns to classify images and eliminating the need for manual feature extraction. They are currently a topic of great interest in machine learning, and have demonstrated excellent performance in classification [10,11].

A spectrogram [12] is a visual representation of the frequencies of a signal as it varies with time. Based on Fourier’s work, it transforms the representation of an audio signal as a function of time into a representation of its frequencies and vice versa. By using discrete Fourier transform algorithms, it is possible to create an image corresponding to the presence of various frequencies in an audio signal across time. Thus, a spectrogram is a picture of a sound and it shows the frequencies that make up the sound, from low to high, and how they change over time, from left to right.

A spectrogram basically provides a two-dimensional graph, with a third dimension represented by colors. Time runs from left (oldest) to right (youngest) along the horizontal axis. The vertical axis represents frequency, which can also be thought of as pitch or tone, with the lowest frequencies at the bottom and the highest frequencies at the top. The amplitude (or energy or loudness) of a frequency at a particular time is represented by the third dimension, color, with dark blues corresponding to low amplitudes and brighter colors up through red corresponding to progressively stronger (or louder) amplitudes.

A mel-spectrogram [12] is a spectrogram where the frequencies are converted to the mel scale. The mel scale is the result of some nonlinear transformation of the frequency scale. It is constructed such that sounds of equal distance from each other on the mel scale also “sound” to humans as if they are equal in distance from one another. Thus, a typical spectrogram uses a linear frequency scaling, so each frequency bin is spaced the equal number of Hertz apart. The mel-frequency scale is a quasi-logarithmic spacing roughly resembling the resolution of the human auditory system.

The CNN models have been combined with sound spectrograms or mel-spectrograms for the generation of classification models [12,13,14,15,16,17]. Specifically, in [13], CNN and clustering techniques are combined to generate a dissimilarity space used to train an SVM for automated audio classification—in this case, audios of birds and cats. Ref [14] uses CNNs to identify healthy subjects using spectrograms of electrocardiograms. Ref [15] employs CNN with three audio attribute extraction techniques (mel-spectrogram, Mel Frequency Cepstral Coefficient and Log-Mel) in order to classify three datasets of environmental sounds. Ref [16] uses three types of time-frequency representation (mel-spectrogram, harmonic-component-based spectrogram, and percussive-component-based spectrogram) in conjunction with CNNs to characterize different acoustic components of birds (43 different species). Ref [17] employs a concatenated spectrogram as input features for the classification of environmental sounds. The authors state that the concatenated spectrogram feature increases the richness of features compared with a single spectrogram.

### 1.3. Previous Work

It is possible to find two approaches from other research groups to the classification of anuran sounds with CNNs in the literature [18,19]. The first one [18] researches the use of convolutional neural networks with Mel-Frequency Cepstral Coefficients (MFCCs) as input for the classification of anuran sounds. It uses a database of 7784 anuran sounds with the following anuran families: Leptodactylidae, Dendrobatidae, Hylidae, and Bufonidae. Two binary CNNs are designed, achieving 91.41% accuracy in identifying Leptodactylidae family members from the rest and 99.14% accuracy for the identification of *Adenomera Hylaedactyla* species. The paper [19] uses two deep-learning methods that apply convolutional neural networks to anuran classification. About 200 calls from 15 frog species from a mixture of web-based collections and field recordings were used for the design. A deep-learning algorithm was used to find the most important features for classification. A 77% classification accuracy was obtained for the dataset of 15 frog species.

In the context of the automatic classification of anuran sounds using other algorithms, a set of works [20,21,22,23] has been published by our research group at the University of Seville. In [20], the MPEG-7 standard was deeply analyzed in order to provide standard acoustic descriptors that could characterize animal sound patterns. The conclusions of this work demonstrate the advantages of the use of this standard in terms of scalability in the wireless acoustic sensor network.

In [21], a set of classifiers based on different algorithms was proposed. The classifier with the best results is based on decision trees, obtaining an overall accuracy of 87.30%. In this work, frames were analyzed without taking into account their relationship with their predecessor and successor. The various classifiers were also compared with a sequential classifier (hidden Markov model).

In [22], a study to analyze the improvements in classification that can be provided by the application of classifiers that consider methods different from those used in [21] (MPEG-7) was proposed for the extraction of characteristics of an audio frame. Two were proposed: energy of filter banks and frequency coefficients of mel. The best results, with an overall accuracy of 94.85%, were obtained using the MFCC (optimal) characteristics and a Bayesian classifier.

In [23], a new algorithm with the same advantages of the classifiers used in [22] was proposed that adds an additional processing that not only considers the label of the classifier that is assigned to each frame, but also considers a score that determines whether this frame belongs to one class or another. To do this, the researchers analyzed the spectrograms of each sound by identifying the regions of interest (ROI) containing anuran sounds. With this approach, the use of two classifiers was identified as the optimal method: minimum distance and decision tree. The results of accuracy in the classification are 97.35%.

In the above-described works by our group [20,21,22,23], a database of real sounds of anurans was used for training the models. This database was provided by the National Natural History Museum (Museo Nacional de Ciencias Naturales) [24]. The sounds were recorded in five different locations, four in Spain and one in Portugal, using a Sennheiser ME80 microphone. They were sampled at 44.1 kHz. The sample included sounds from 2 anuran species: the *epidalea calamita* (natterjack toad) and the *alytes obstetricans*. Three types of vocalization were distinguished for the *epidalea calamita*: standard, chorus, and amplexus. Two types of vocalization were distinguished for *alytes obstetricans:* standard and distress call. This same database was used for training the classification system of the present work. In this way, it should also be possible to compare the results of the CNN against the models used in previous studies.

### 1.4. Research Objectives

The research objectives of this paper are twofold:The implementation of a cyber-physical system to optimize the execution of CNN algorithms: audio wave processing based on CNN [10] has been frequently proposed in the literature, as shown in Section 1.2. However, an optimal execution model must be considered in order to maximize flexibility and adaptability of the algorithms, and to minimize the execution costs (power consumption and time response). Thus, a first research objective is to propose the implementation of a cyber-physical system (CPS) to reach these goals. Figure 3 represents the main components of the CPS proposal. The physical system is composed of IoT nodes that create an ad hoc network. These nodes are equipped with microphones adapted for natural environments. Every node captures the soundscape of its surroundings. The audio signal is converted to a mel-spectrogram [12], which is processed using a set of trained CNNs. This data fusion procedure reduces the size of the information to be sent through the network. The nodes are powered by solar cells and have a wireless interface. The physical system is modeled in a set of servers following the paradigm of digital twins [25]. This constitutes the cyber-physical part of the system. Every physical node has a model (digital twin) that can interact with other models, creating a cooperative network. At this point, the CNNs are trained with the audio data recorded previously by the physical nodes. There are as many CNNs as species or biological acoustic targets being studied. Once a CNN is trained, its structure, its topology, and its weights are sent to the corresponding physical node, where it will be run. The execution of every CNN offers events of identification of a given species. The entire system can be monitored by scientists with a user-friendly interface.

The design of a CNN classification system for anuran sounds: For this second research, the different types of anuran vocalizations that our research group has worked on in the past (see Section 1.3) would be used as training set. Obtaining good results would allow a demonstration of the feasibility of a CNN-based design for the classification of species sounds and, in general, for biological acoustic targets. At the same time, having previous work in the classification of this type of sounds allows us to compare the results of the CNN with other techniques. Thus, although the results of the most recent of the previous works [23] were satisfactory, there were two areas in need of improvement. The first one focused on creating a system that performs an even finer classification of anuran sounds both in terms of accuracy and in number of classified sounds. A second objective was to remove the previous manual preprocessing of the audio signals carried out in [23]. This would speed up the classification process and save costs. The different stages in the designed classification process are shown in Figure 4.

Firstly, the environmental audios are captured with a microphone and stored in WAV or MP3 format. Subsequently, the mel-spectrogram is generated for the audio file and stored in JPEG format. Finally, the CNN carries out the detection and classification of the anuran vocalization from the mel-spectrogram.

This paper is organized as follows. Section 2 gives the details of the creation of the cyber-physical system and the CNN classification system. The results of the evaluation of the system are described in Section 3. Section 4 offers a discussion about the results and some conclusions are provided in Section 4.

## 2. Material and Methods

### 2.1. Cyber-Physical System

As was seen in the previous section, most of the approaches to environmental monitoring are mainly based on two strategies:Sensors acquire data and send them to a base station from where they are forwarded to a central processing entity. In this scenario, data are sent raw or with basic processing. In order to enhance the management of the communications, data could be aggregated. The main drawback to this approach is the huge data traffic that the communication infrastructure must allow. This feature is especially important in low-bandwidth systems.Sensors acquire data and process them locally. In this scenario, a data fusion algorithm is executed locally in every remote node or in a cluster of nodes that can cooperate among themselves. Only relevant information is sent to sink nodes, which deliver it to supervising and control entities. This sort of architectures reduces the data traffic and consequently the power consumption. The required bandwidth for these nodes can be minimized, allowing low-power wireless sensor network paradigms such as IEEE 802.15.4^TM^ [26], LoRa^®^ (LoRa Alliance, Fremont, CA, USA https://lora-alliance.org, accessed on 19 May 2021) [27], etc. The main drawback of these systems is the necessity of adapting the data fusion algorithm to nodes with processing and power consumption constraints.

The first approach is based on a centralized architecture, as depicted in Figure 5. A central processing node spreads its sensor nodes out following a cluster tree communication infrastructure. Data can flow from source nodes through intermediate nodes (called base stations) that forward the packets toward the sink node. In this paradigm, the nodes that acquire data (e.g., audio waves) from the environment are assumed to be in the physical domain. In Figure 5, this domain is represented in black. Additionally, the nodes that process data usually are located in a data center considered the cyber domain (represented in grey in Figure 5). The main processing tasks are developed in the cyber domain. Typical processing algorithms are based on artificial intelligence techniques such as convolutional neural networks for classification purposes. The construction of this artificial neural network is based on two stages, both carried out in this central entity. In the first stage, the network is trained with the data collected from sensor nodes (audio wave). In this stage, human supervision is needed to identify every output with its correct pattern. In the second stage, the network is run with the trained supervised neural network.

While this approach delivers the control and supervision responsibility to a centralized entity called the sink node (with no hardware constraints), the decision is made remotely from data sensors, which may cause inflexibility due to its fixed structure and the latency of its links. Furthermore, the processing is carried out in a central entity that could create a bottleneck. Possible dysfunctionalities of this entity could cause the system to fail.

The second approach is based on a decentralized architecture, as shown in Figure 6. In this scenario, every source node acquires data and processes them locally. In this case, the CNN is run on the physical domain. There are as many CNNs as there are sensor nodes. Every CNN implements a classification algorithm that provide a short plain text file as result. These files are collected by the sink node in an aggregation scheme. Therefore, the required bandwidth on every link is not considered a challenge. Human supervision must be carried out on every node in order to fit the algorithm to the target under study.

In contrast with both the first and second approach, the cyber-physical systems (CPS) approach combines the execution of the algorithms in both the physical and the cyber domains. Figure 7 shows an example of how a CPS architecture could fulfill the requirements of a classification problem while minimizing the drawbacks found in the prior approaches.

The strategy followed in this approach develops a dual-stage implementation of the CNNs: training carried on in the cyber system and execution over nodes in the physical system. This strategy must be analyzed from the viewpoint of several metrics. Because CPS follows a distributed processing strategy, the communication features among nodes and between physical and cyber systems play an important role in the achievement of the objectives. On one side, the low data rate in wireless links among nodes does not support the transmission of raw audio data through the network. Due to this feature, a possible central CNN execution scheme is discarded. Instead, a local CNN execution scheme must be implemented. Furthermore, before the execution of the CNN, a pre-processing stage is needed: the creation of the mel-spectrogram. Therefore, nodes must execute the CNN using its own processing resources. Two possibilities can be considered: pre-process the audio signal on the central processing unit (CPU) and execute the CNN on the graphical processing unit (GPU); or pre-process the audio signal and execute the CNN on the CPU. In the work proposed here, both the pre-processing of the audio signal and the execution of the CNN are carried out by the CPU using a TensorFlow Lite library [28]. The platform used for the testbeds is based on the System of Chip (SoC) BCM2835, which includes the microprocessor ARM1176JZF-S at 700 MHz—it can reach 1 GHz doing overclocking using GPU VideoCore IV. The communications were implemented using a LoRa^®^ with a data rate of around 5kbps using a spread factor of 7 and a bandwith of 125 kHz [29]. Nodes are provided with a weatherproof microphone and a solar cell for an autonomous power supply.

On the cyber system side, implementation was based on a workstation with an AMD Threadripper Pro 3955WX processor with 32 GB RAM and GPU NVIDIA Quadro P620, running Windows 10 Pro. The communication between the cyber and the physical system was made using the TCP/IP protocol. A Python application was designed to train the CNN based on a selected dataset. This cyber system is responsible for interacting with a final user (biologist) who is responsible for ordering the new trainings of the CNNs and new updates of the trained CNNs on the physical nodes or on their digital twins to carry out simulations or cooperative classifications.

The CPS is being tested at the Doñana Biological Reserve ICTS (DBR) and the Higher Technical School of Computer Engineering (ETSII), both located in the south of Spain with a separation of 60 km. Figure 8 depicts the actual location of every part of the system. The physical system is composed of 9 IoT nodes spread out over Ojillo Lake, a natural environment in DBR. These nodes create a spanning tree with a base station as a root. The communication among nodes is carried out by LoRa^®^. A node with two interfaces (Ethernet and LoRa^®^) performs a base station role. This node communicates with the cyber system through fiber optics.

### 2.2. CNN Classification System

In order to carry out the design of the CNN classification system, a set of stages were followed. These stages are represented in Figure 9 and described in the following subsections.

Each of the stages was programmed with Python using the Keras library [30] for CNNs.

#### 2.2.1. Data Augmentation

In order to generate a more robust training set, an augmentation process was carried out using the sound database. When one deals with CNN and images, data augmentation consists of extending the set of images from their transformations (rotations, size modifications, etc.). For the purpose of the present work on sounds, the transformations must be made on the audios, and later these transformations will be reflected in the mel-spectrogram images.

In this way, for each of the 865 audio samples in the database, 10 additional audios were generated:2 files with two types of white noise added to the audio.4 files with signal time shifts (1, 1.25, 1.75, and 2 seconds, respectively).4 files with modifications in the amplitude of the signal: 2 with amplifications of 20% and 40%, and 2 with attenuations of 20% and 40%. Specifically, dynamic compression [31] was used for this processing. It reduces the volume of loud sounds and amplifies (a percentage) the quiet ones. The aim was to generate new signals with the background sound enhanced and the sound of the anuran reduced.

These transformations were carried out so that that the system could detect the anurans in a noisy environment, as well as at different distances from the recording microphone. With the previous transformations, the 865 audios were converted to a total of 9515 audios.

After the data augmentation stage, the distribution of 5-second-long samples among the different classes is detailed in Table 1.

In the works cited above [20,21,22,23], it was assumed that due to the similarity of the chorus in *Epidalea calamita* with respect to standard vocalization, as well as the low number of chorus vocalization samples, it was not possible to train a system to distinguish between the two. As a contribution to that work, in the present work a CNN was generated for the classification of the 5 classes registered in the database. The objective was to demonstrate that the power of this type of CNN model allowed satisfactory results to be obtained in all vocalizations.

#### 2.2.2. Mel-Spectrogram Generation

The generation of a mel-spectrogram for each audio file was perfectly adapted to the classification objective of the system. The WAV files were passed through filter banks to obtain the mel-spectrogram. Each audio sample was converted to a shape of 128 – 435, indicating 128 filter banks used and 435 time steps per audio sample.

Figure 10 shows the audio corresponding to a sample with an amplexus call of *Epidalea calamita*.

As can be seen in the figure, the anuran call is located in the interval (2.5, 3.5 s) of the sample. The mel-spectrogram allows identification, in the image, of both the location of the call and the range of frequencies and amplitudes of the same.

#### 2.2.3. Design and Training

As detailed in the introduction, a CNN is a type of neural network basically composed of convolution layers and pooling layers. For the design of this work, the convolution layers extract features from the mel-spectrograms, preserving the relationship between pixels by learning image features using small squares of input data. Thus, the role of a convolution layer is to detect call or vocalization features at different positions from the input feature maps with these learnable filters. The pooling layers are used to reduce the dimensionality of the previous layers. For this work, max pooling was used. This type of pooling calculates the maximum value in each patch of each feature map.

For the classification system presented in the paper, the best results were found with CNNs of 3 convolution layers and 3 pooling layers. A flattened layer and a ReLU (Rectified Linear Unit) layer were added. The flattened layer simply converts the pooled feature map (represented as a numeric matrix) into a column (represented as a vector). ReLU [32] stands for rectified linear unit and is a nonlinear operation defined by max (zero input). The purpose of applying the rectifier function is to increase the nonlinearity in the images (the images are naturally nonlinear). Finally, softmax functions with cross-entropy [32] were applied in the output layer. This turns the output vector of real values into a vector that sums to 1 so that the results can be interpreted as probabilities. The number of output filters per convolution layer is 32 for the first one, 64 for the second one, and 128 for the final one. A kernel size (height and width of the convolution window) of 3 × 3 is used on each convolution layer. The fully connected layer (FC) consists of 512 neurons and the softmax layer with the number of units for classification (one for each type of sound). Figure 11 shows the structure of the CNN.

For network training, 80% of the samples were used as training samples, and 20% as tests. This strategy is widely used to enhance the generalization of a machine learning model and prevent overfitting. A balanced sampling was carried out for the test set. For it, one audio sample (with its corresponding data augmentation samples) out of 5 was selected for each type of anuran call. Thus, 7623 samples were used for training and 1892 for testing.

As mentioned in the data augmentation section above, a CNN was first trained for the 4 sound classes covered in the classification model developed in previous work [20,21,22,23]. Later, a second CNN was trained to expand the classification system to the total of the 5 classes detailed in Table 1.

For the first network, the best results were obtained with training in only 8 epochs (number of passes of the entire training dataset) and a batch size (number of samples utilized in one training iteration) equal to 32. For the second network, the best training was achieved with 6 epochs and, again, a batch size equal to 32. For both networks, the initial learning rate [33] with 0.001 and the “adam” optimizer [34] were the parameters used in the training process.

## 3. Results

### 3.1. CPS Performance

Several factors were considered in evaluating the CPS. First, at the level of data transport communication, a comparison between two paradigms was done. The central paradigm is based on sending the audio record from an IoT node to its digital twin at the cyber system side, where it is processed. The distributed paradigm is based on processing the audio record in the IoT node and sending the obtained result to its digital twin, where it is delivered to the central processing entity. Considering a 5-second audio register recorded with a sample rate of 44.1 kHz and 32 bits per sample with mono configuration, the resulting file (WAV) has a size of 434 kB. For this, it was assumed that a single radio link (1 kbps) exists and the minimum time needed to transport the audio file from the physical system to the cyber system is 57.8 seconds. In this scheme, the time associated with data communication at the fiber optics was negligible in comparison with the data communication at the radio link. As a result, sending 5 seconds of audio record needs more than 10 times more time. Consequently, no real streaming could be done. A distributed processing is needed. In this sense, two times have to be considered at the physical nodes: pre-processing time (mel-spectrogram representation) and CNN execution. Considering the same previously created WAV file, the time to compute the mel-spectrogram was measured in 120 ms and the later CNN execution, using TensorFlow Lite library, took 160 ms. Finally, the time for sending the CNN results of 5 seconds audio record analysis (70 B) implies 0.55 seconds. As a result, the total amount of the time needed in the distributed paradigm is around 300 ms. In conclusion, it is possible to do an audio analysis in real time. The main handicap of this approach is that if any change in the CNN is needed, the network topology, weights, and structure have to be transmitted through the radio links. A common neural network size using TensorFlow Lite is about 5 MB. With a 1kbps bandwith, the time to deliver this network from the cyber system to a remote IoT node takes 11.4 hours. This time, although significantly high, corresponds to a procedure that is not done frequently and can be done in the background.

Second, at the level of boundary conditions, the temperature of SoC at the IoT has been measured while executing the CNN. Figure 12 shows a thermal photo that identifies the temperature at the SoC (48.8 °C). This shows that the selected platform is suitable for executing this kind of processing. This testbed was carried out at 25 °C (spring and fall); additional tests should be considered at higher temperatures (summer).

Third, at the level of power consumption, the execution of the CNN was tested using a precision multimeter. Figure 13 depicts the level of intensity required by the IoT platform when the CNN begins its execution. The solar cell installed in the remote location where the node is working has been dimensioned according to the IoT power consumption.

Fourth, at the level of resources, the application that runs the CNN takes 16.4% of the CPU capacity and 8.7% of RAM. These results let us conclude that the available computing resources are adequate for running this sort of application.

### 3.2. CNN Models

The overall accuracy results obtained for the first network, which classified the sounds in four classes, were 96.34% for training and 97.53% for the test data. The confusion matrix for the test data in this network is detailed in Table 2.

The dispersion of the error between the four classes is low, standing between 1.21% for the *Alytes obstetricans* class (11 errors versus 913 audios) and 4.9% for the *Epidalea calamita* class with amplexus vocalization (7 errors versus 143 audios).

For CNN, which classifies the 5 types of sound detailed in Table 1, a 94.54% success rate was obtained for the training data, and 96.2% for the test data. Table 3 shows the confusion matrix obtained for the test data.

Table 3 shows how the errors are distributed among the 5 different classes. The percentage of error for all classes is in the range between 1.01% and 4.78%, except for the *Epidalea calamita* class with amplexus vocalization, which has a 9.09% error (13 errors versus 143 audios). On the other hand, only one error was made in the *Alytes obstetricans* class with distress calls. Thus, the network classification results can be considered good and robust at the same time. The accuracy results for this network with respect to the first network decrease slightly, but they are good enough to show that the CNN is capable of distinguishing between standard and amplexus vocalization for *Epidalea calamita*.

## 4. Discussion and Conclusions

Regarding the CNN-based classification system, as shown in Section 3, the accuracy results are excellent and robust for the set of anuran sounds. Specifically, in the CNN based on 4 sound classes, the overall accuracy result obtained (97.53%) involved a small improvement with respect to the best results (97.35%) of the most recent paper [23] from the previous work on this sample set [20,21,22,23]. Thus, if Table 2 of Section 3 is compared with Table 7 of [23], it is possible to observe improvements in the accuracy results of the classes *Epidalea calamita* and *Alytes obstetricans* with standard vocalization. The result is only worse in the *Alytes obstetricans* class with distress calls, in which a classification error occurs in one of the 44 samples. Although the improvement in accuracy is not significant, in [23] a manual labeling of the sounds is required, which adds a cost to the process. In the classification system in the present work, taking advantage of the capabilities of deep learning, this human help is not necessary, which is a significant contribution. Furthermore, previous works could not distinguish between chorus and standard vocalization for *Epidalea calamita*. The classification system based on the second CNN makes it possible to obtain very good results (96.2%) when adding this new class. This involves a greater level of detail in sound classification and implies a major improvement with respect to the previous work cited above. This classification system, designed for anuran sounds, demonstrates the feasibility of a CNN-based design for the classification of biological acoustic targets. In this way, the first objective of those proposed in the paper (see Section 1.4) was completed. On the other hand, although this paper has focused on the use of a CPS as an anuran classification tool, it could also be used to analyze biodiversity indices in the natural environment, using the proper databases for CNN training.

Regarding the CPS, the proposed scheme provides a promising chance for biologists and engineers to collaborate in complex scenarios, minimizing the supervision of experts and, consequently, the cost in human hours. Doubtless, the CPS could be implemented with less costly devices. Consequently, extensive natural areas could be included for analysis in monitoring applications.

Given the results reported here, the authors conclude that the CPS design constitutes a novel, flexible, and powerful PAM system. Overall, the use of CNNs in IoT nodes could be used in the classification stage as a fusion technique that minimizes the data to be transported and maximizes the information exchanged. Thus, the second of the research objectives of the work (see Section 1.4) was completed.

As a future line of work, CNNs can be engaged not only in classification but also in feature extraction. In the present work, CNNs were used with classification as the main objective. However, audio data could contain hidden information that has not been considered with this approach. A deeper insight in an unsupervised execution training may provide new interesting conclusions regarding the process dynamics.

## Figures and Tables

**Figure 1 sensors-21-03655-f001:**
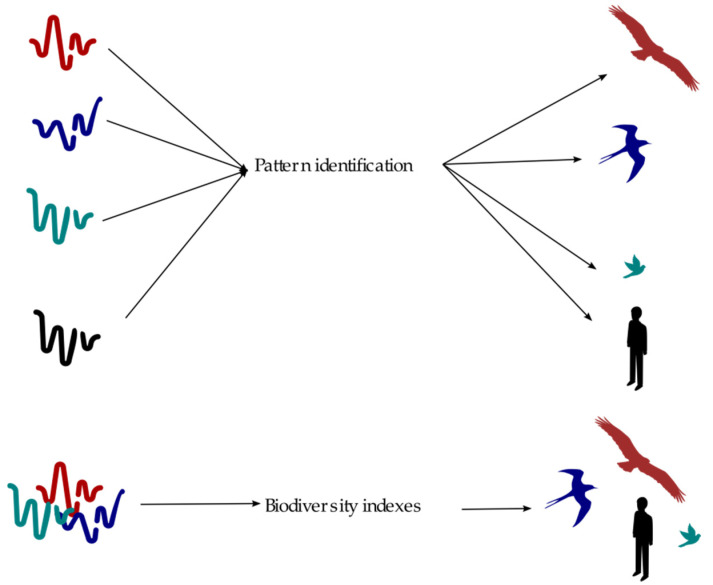
Stages in the CPS of SCENA-RBD project.

**Figure 2 sensors-21-03655-f002:**
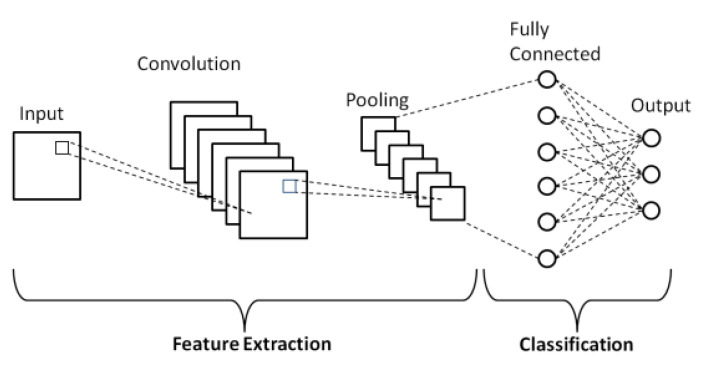
CNN Architecture.

**Figure 3 sensors-21-03655-f003:**
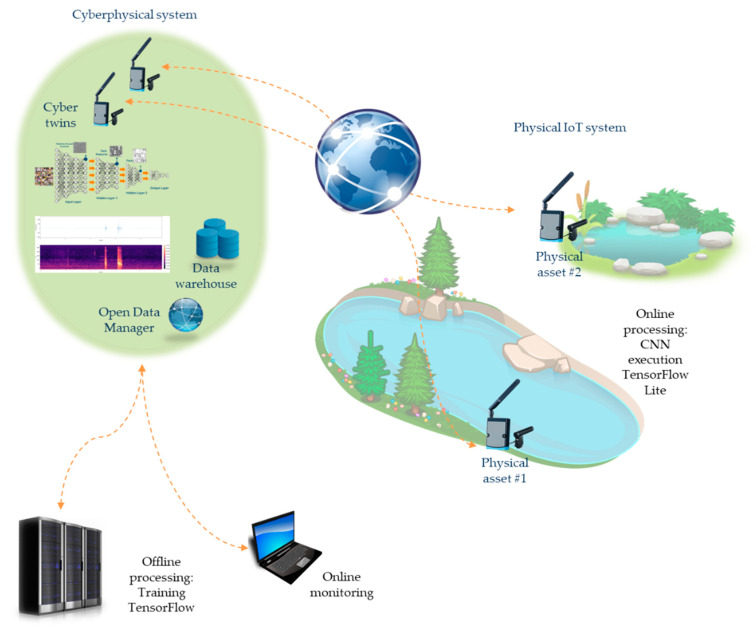
Main components of the CPS.

**Figure 4 sensors-21-03655-f004:**
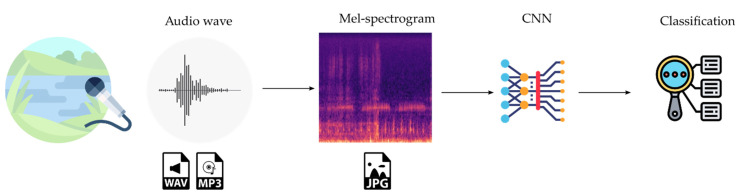
Stages in the CNN classification system.

**Figure 5 sensors-21-03655-f005:**
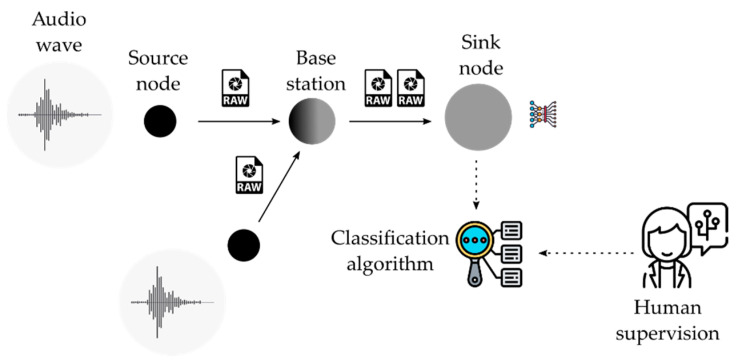
Centralized architecture.

**Figure 6 sensors-21-03655-f006:**
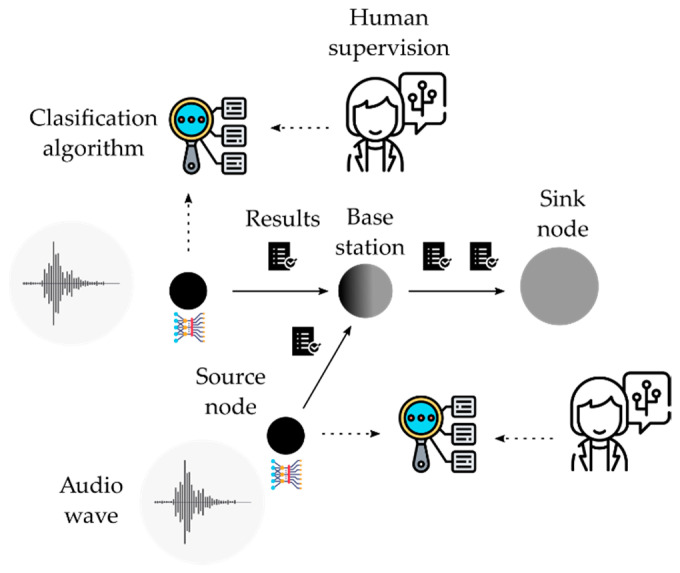
Decentralized architecture.

**Figure 7 sensors-21-03655-f007:**
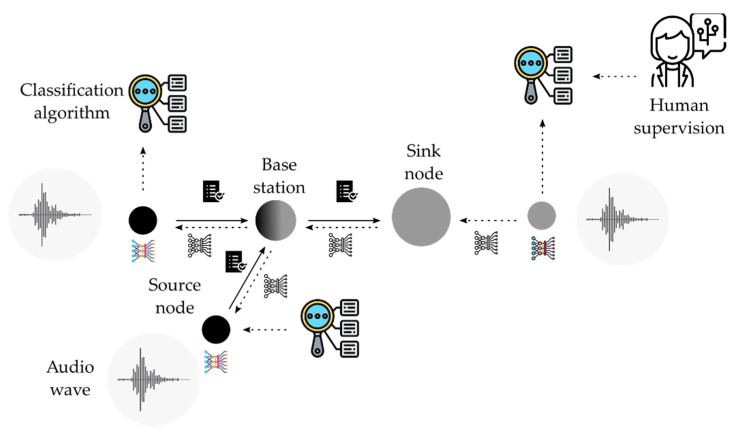
Cyber-Physical System architecture.

**Figure 8 sensors-21-03655-f008:**
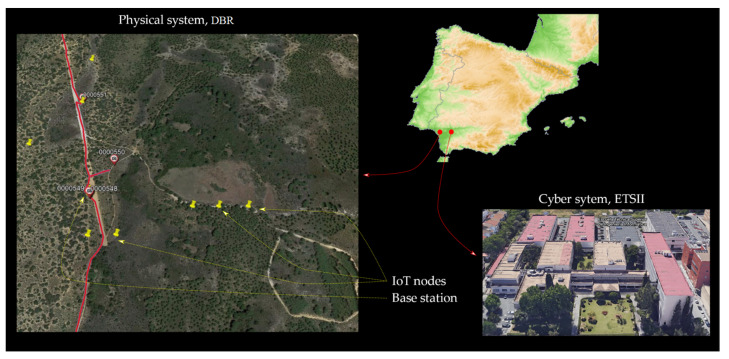
CPS topology. Physical system, DBR, Huelva, Spain; Cyber system, ETSII, Sevilla, Spain.

**Figure 9 sensors-21-03655-f009:**
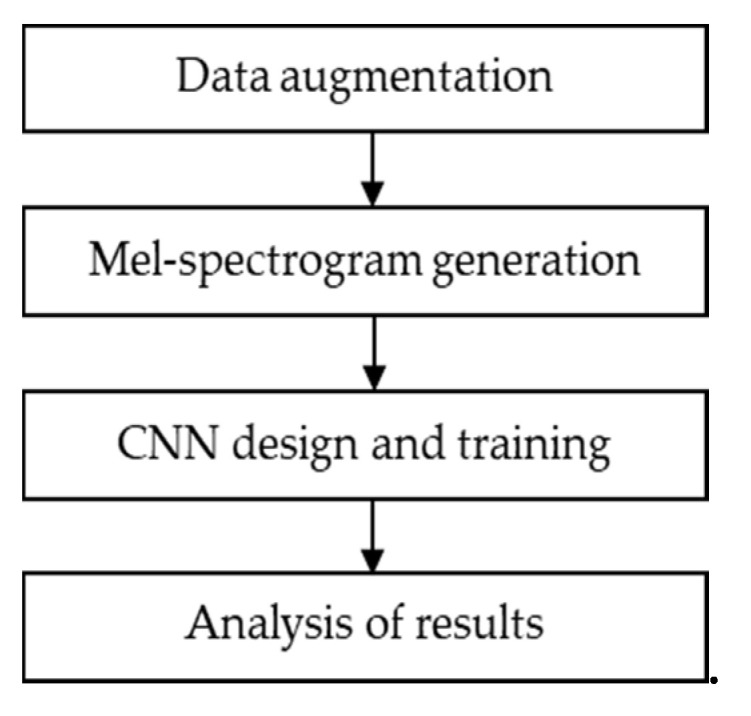
Stages in the classification system.

**Figure 10 sensors-21-03655-f010:**
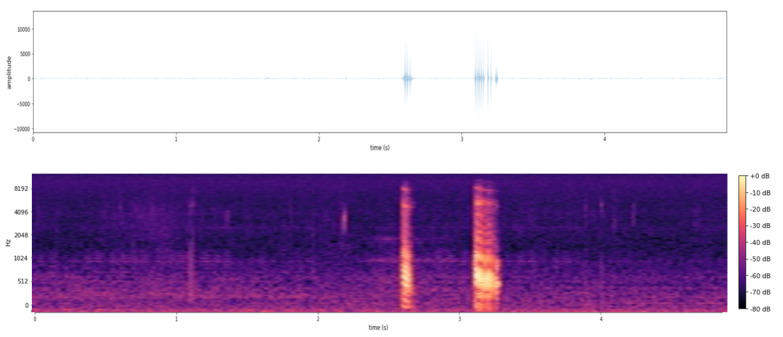
Example of anuran call: waveform (up) and mel-spectrogram (down).

**Figure 11 sensors-21-03655-f011:**
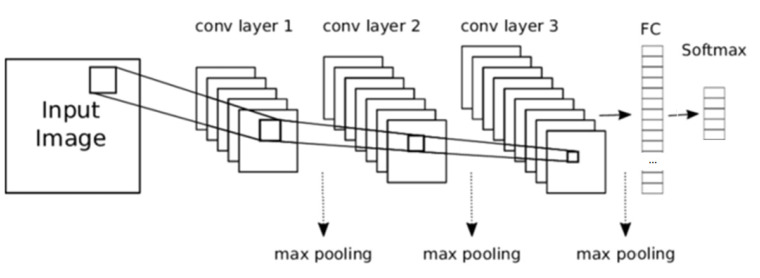
Structure of the designed CNN.

**Figure 12 sensors-21-03655-f012:**
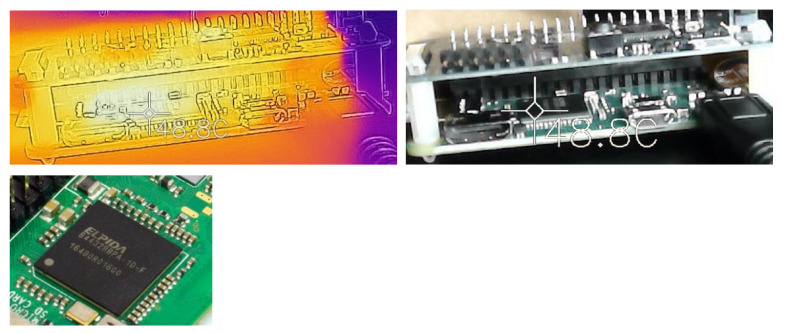
Temperature at the SoC of the IoT node executing the CNN.

**Figure 13 sensors-21-03655-f013:**
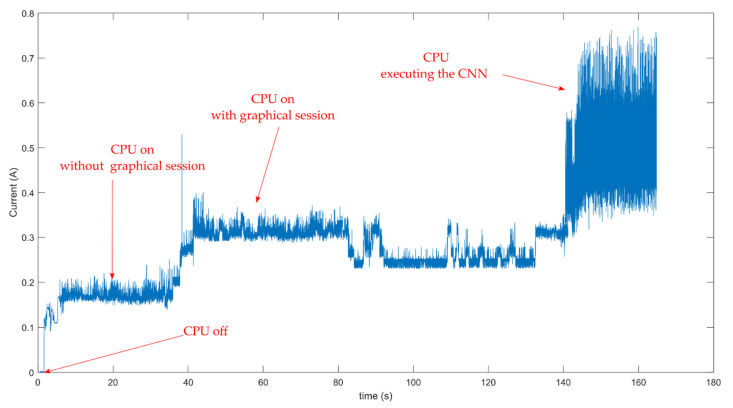
Power consumption of the IoT executing the CNN.

**Table 1 sensors-21-03655-t001:** Sound collections.

Type	Vocalization	Number of Original Samples	Number of Augmented Samples
*Epidalea calamita*	standard	293	3223
*Epidalea calamita*	chorus	74	814
*Epidalea calamita*	amplexus	63	693
*Alytes obstetricans*	standard	419	4609
*Alytes obstetricans*	distress call	16	176

**Table 2 sensors-21-03655-t002:** Confusion matrix for the first CNN (4 classes).

		Predicted Values			
		*Ep. cal. st&ch*	*Ep.cal. amplexus*	*Al. obs. standard*	*Al. obs. distress*
**Actual values**	***Ep. cal. st&ch***	96.51% (775)	3.49% (28)	0	0
	***Ep.cal. amplexus***	4.9% (7)	95.10% (136)	0	0
	***Al. obs. standard***	1.21% (11)	0	98.79% (902)	0
	***Al. obs. distress***	2.28% (1)	0	0	97.72% (43)

**Table 3 sensors-21-03655-t003:** Confusion matrix for the first CNN (5 classes).

		Predicted Values				
		*Ep. cal. standard*	*Ep. cal. chorus*	*Ep. cal. amplexus*	*Al. obs. standard*	*Al. obs. distress*
**Actual values**	***Ep. cal. standard***	94.27% (560)	1.01% (6)	4.71% (28)	0	0
	***Ep. cal. chorus***	4.78% (10)	95.22% (199)	0	0	0
	***Ep. cal. amplexus***	9.09% (13)	0	90.91% (130)	0	0
	***Al. obs. standard***	1.2% (11)	0	0	98.8% (902)	0
	***Al. obs. distress***	2.28% (1)	0	0	0	97.72% (43)

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
