# Peer review of "Cyber-Physical System for Environmental Monitoring Based on Deep Learning"

_sensors, 2021, doi:10.3390/s21113655_

Round 1
Reviewer 1 Report
The paper is well written and it gives insights for processing cyber-physical systems, in particular, show excellent results for CNN-based classification systems for a set of anuran sounds. The results are outstanding, and my concerns are focused only on:
Section1.3 Previous work is strongly centered on the work of the research group. In my appreciation, this paper will improve if this section includes the overall development in the area instead of the work made by the research group.
Section 1.4 Researches objectives. These objectives are a bit generic. Should be better to include only one objective for this paper, and show in the conclusions that this objective was achieved.
Not all the figures and tables fit with the document width (for example, fig 13). What is the source of the image in Fig 8?.
Reviewer 2 Report
Revise the manuscript along the guidelines given in the attached file.

Reviewer 3 Report
This paper demonstrates the feasibility of a Cyber-physical systems (CPS) for monitoring and classifying biological acoustic targets using Internet of Things (IoT) devices. The classification system is based on convolutional neural networks (CNNs) and much of it is implemented locally at the IoT nodes which is very advantageous in the context of IoT. The classification results obtained are excellent (97.53% overall accuracy) and can be considered a very promising use of the system for classifying other biological acoustic targets as well as analyzing biodiversity indices in the natural environment.
In general the paper is well-written and easy to follow. The authors have a number of publications in the area of automatic classification of anuran sounds. Also, as mentioned by the authors, audio wave processing based on CNN has been frequently proposed in the related literature. The main contribution of this work lies in the optimal configuration of a CPS in terms of maximizing flexibility and adaptability of the CNN algorithms and to minimize the execution costs (power consumption and time response). Through the provided analysis and results, the authors successfully demonstrate that the execution of this type of CPS, involving low-cost IoT devices and reduced computing resources, are indeed a feasible solution towards monitoring the natural environment.
Round 2
Reviewer 2 Report
Most of the review questions and have answered by the authors and necessary improvements are made in the manuscript.
Kindly check the manuscript for finer details and typos.